# Multifaceted Hybrid Carbon Fibers: Applications in Renewables, Sensing and Tissue Engineering

**Chandreyee Manas Das [1],[†], Lixing Kang [1],[†] , Guang Yang [1], Dan Tian [2],* and Ken-Tye Yong [1],***

1   School of Electrical and Electronic Engineering, Nanyang Technological University, 50 Nanyang Avenue, Singapore 639798, Singapore; CHANDREY001@e.ntu.edu.sg (C.M.D.); lxkang@ntu.edu.sg (L.K.); YANG0389@e.ntu.edu.sg (G.Y.)
2   College of Materials Science and Engineering, Nanjing Forestry University, Nanjing 210037, China
*   Correspondence: tiandan@njfu.edu.cn (D.T.); ktyong@ntu.edu.sg (K.-T.Y.)
†   The authors contributed equally to this work.

**Abstract:** The field of material science is continually evolving with first-class discoveries of new nanomaterials. The element carbon is ubiquitous in nature. Due to its valency, it can exist in various forms, also known as allotropes, like diamond, graphite, one-dimensional (1D) carbon nanotube (CNT), carbon fiber (CF) and two-dimensional (2D) graphene. Carbon nano fiber (CNF) is another such material that falls within the category of CF. With much smaller diameters (around hundreds of nanometers) and lengths in microns, CNFs have higher aspect (length to diameter) ratios than CNTs. Because of their unique properties like high electrical and thermal conductivity, CNFs can be applied to many matrices like elastomers, thermoplastics, ceramics and metals. Owing to their outstanding mechanical properties, they can be used as reinforcements that can enhance the tensile and compressive strain limits of the base material. Thus, in this short review, we take a look into the dexterous characteristics of CF and CNF, where they have been hybridized with different materials, and delve deeply into some of the recent applications and advancements of these hybrid fiber systems in the fields of sensing, tissue engineering and modification of renewable devices since favorable mechanical and electrical properties of the CFs and CNFs like high tensile strength and electrical conductivity lead to enhanced device performance.

**Keywords:** carbon nano fiber; sensing; tissue engineering; renewables

## 1. Introduction

The science behind materials plays an important role in almost every aspect of engineering, medicine and industry. To be more precise, the dimension of these materials can completely alter the way they interact with other physical aspects like electromagnetic radiation. For instance, when it comes to bulk materials, these do not display any spectacular phenomenon upon interaction with light. However, when we go to the nano level in terms of size, the material property changes entirely and can give rise to many novel physical principles like surface effect, quantum size effect, quantum tunneling effect, dielectric confinement effect and many more. The multifarious carbon nano fibers (CNF) have excellent thermal, mechanical and electrical properties and they are being utilized in a variety of fields like aerospace, transportation, civil engineering and green technology. However, they also possess some limitations like diminished specific surface area, lipophobic surface and low chemical activity. Thus, to take the maximum advantage as well as reduce the physical restrictions, many researchers make use of hybrid materials.

CNFs are mainly prepared by two methods namely the catalytic thermal chemical vapor deposition (CVD) and the electrospinning process followed by heat treatment. There are two types of CNFs

that can be prepared by the CVD method: cup-stacked CNF and platelet CNF. In the CVD method many types of metals that can dissolve carbon to form metal carbide are used like Iron, Nickel, Cobalt, Chromium, Vanadium. To obtain carbon sources in the range of 700 K to 1200 K, Molybdenum, Methane, Carbon Monoxide, Synthesis gas ($H_2$/CO), Ethyne or Ethene are used. In the electrospinning process, polymer nanofibers are required as precursors. The properties of the obtained CNF depend on the polymer solution used and the processing parameters. Polyacrylonitrile (PAN), Poly (vinyl alcohol) (PVA), Polyimides (PIs), Polybenzimidazole (PBI), Poly (vinylidene fluoride) (PVDF), Phenolic resin and lignin are the common polymers that are used. After successfully preparing the polymer nanofibers, heat treatment is used to carbonize the polymer nanofibers to form CNFs. The shape, porosity, diameter and other structural characteristics are governed by the physical conditions of the heat treatment process like temperature and pressure [1].

The main physical properties that make CNF utility unique in sensors and other devices is due to their electrical, thermal and mechanical behavior. The addition of CNF in polymers enhances its mechanical properties. In general, by just a minute addition of CNF, the resistance to fracture is greatly enhanced. The addition of 0.5 wt% and 1.0 wt% CNF in epoxy enhances its fracture resistance by 66% and 78% respectively. In another application, by 4.0 wt% addition of CNF in thermal-plastic polyurethane (TPU), the tensile strength increased by 49% as compared to neat TPU. However, increased addition of CNF can be counterintuitive as it can result in void formation and other defects due to development of bundles of CNF that lead to stress concentration and easy fatigue resulting in premature failure. Coming to the thermal properties, the addition of CNF in the matrix material can enhance its thermal conductivity leading to better heat dissipation and reduced chances of thermal failure. Many models have been developed for prediction of thermal conductivity of CNF composite, like the Maxwell model and the series/parallel model, as a function of individual thermal conductivities of the matrix and filler materials. The enhanced electrical conductivity of CNF composites is due to the tunneling effect, where conductive pathways are developed in the matrix material as a result of the addition of CNF. Many theories and models have been developed to ascertain the reason behind the enhanced electrical conductivity of these composites [1].

CNT and Graphene have been used in many applications of bio-sensing, therapeutics and flexible electronics because of their favorable electrical and mechanical characteristics. However, CNF has its own benefits and is unique in its own way. They are discontinuous and highly graphitic in nature and can be easily blended with polymer processing techniques because of which the development of CNF composites is considerably less complicated. They can be hybridized easily with a wide range of matrix materials including thermoplastics, thermosets, elastomers and others. With just minute additions of CNF, the mechanical, thermal and electrical properties of the matrix material can be significantly improved. Hence, it is quite cost effective to manufacture CNF composites and they can be commercialized easily.

The structure of the review is divided into three parts. In the coming sections we discuss the applications of CF and CNF in renewable energy sector in Section 2, especially concentrating on batteries, supercapacitors, solar cells and fuel cells. Section 3 will deal with sensing applications of CF and CNF and Section 4 will be about usage of CF and CNF in tissue engineering and finally we end with a concluding note in Section 5.

Below we take a brief look into some efforts made by researchers in developing CF composites and reinforced polymers with enhanced mechanical, thermal and electrical properties.

Carbon fiber (CF) composites display poor interfacial adhesion due to which stresses are not transferred well from the matrix to the reinforcing fibers. Thus, these composites are prone to interfacial failures. Semitekolos et al. used poly methacrylic acid (PMAA) to modify carbon fiber (CF) fabric in order to attain better fiber–matrix interfacial strength [2]. With strong hydrogen bonding between the respective carboxyl and hydroxyl groups of PMAA and epoxy resin, an interlayer was created that showed maximum enhancement in Interlaminar Shear Strength (ILSS). Structural defects pose significant hindrance in the path of enhancing mechanical properties of CFs. In order to improve the

electrical conductivity and mechanical strength of these fibers, Sui et al. used Polyacrylonitrile (PAN) nanofibers embedded with 0–20 wt% Multi-walled Carbon Nanotubes (MWCNT) to generate hybrid nano-scale CFs with the help of electrospinning [3]. They observed significant improvement in electrical conductivity of about 26 Scm$^{-1}$ even with 3 wt% addition of MWCNT. In another interesting application of hybrid CNF, Lui et al. used CNF as a lubricating filler [4]. They modified high strength glass fabric (HSGF)/phenolic laminates with 1 to 3% CNF and enhanced the ILSS. In addition to high modulus and strength, because of the self-lubricating properties of CNF, the composite material displayed superior resistivity towards corrosion in water-based environments. Ulus et al. prepared hybrid nanocomposites made of CF, Boron Nitride Nano Particles (BNNP) and CNT [5]. The addition of BNNP and CNT enhanced the tensile, flexural and shear strengths of epoxy resin and CF. Scanning Electron Microscopy (SEM) images showed minimum damage and maximum improvement in mechanical properties for BNNP–CNT hybrid nanocomposites. The enhancements in bending stiffness and shear strengths for BNNP–CNT-Epoxy/CF were 38.5% and 90%, respectively, in comparison to plain CF/Epoxy composites. Sui et al. prepared a PAN–nano CF composite by multi-step hot stretching that displayed higher mechanical strength, was lightweight and had less structural deformations [6]. For polyetherimide composite membranes containing 1 wt% nano CF, there were 21% and 60% improvements in tensile strength and Young's modulus, respectively.

Gabr et al. added nano-clay as filler material into CF polymer composites in order to improve the strength [7]. Using dynamic mechanical analysis (DMA) and SEM they studied CF/compatibilized polypropylene (PPc)/organoclay composites and found that at 3% of the filler material, mode I initiation and propagation interlaminar fracture toughness improved by 64% and 67%, respectively. The SEM images showed that the fibers pulled themselves at the tip during initiation delamination. Additionally, Zhang et al. prepared a ternary biocomposite comprising of nano-hydroxyapatite/polyamide66 (HA/PA) and CF [8]. They observed that CF bonded well with the HA/PA matrix. Higher wt% of CF lead to better enhancements in compressive, bending and tensile strengths that ranged in between 116–212 MPa, 89–138 MPa and 109–181 MPa, respectively. The HA/PA/CF composite also showed high cytocompatibility towards MG-63 cells and thus they demonstrated that the composite can also have potential applications as bone repair materials.

CFs show high thermal and electrical conductivity. However, there are certain applications that simultaneously require good thermal conductivity and electrical insulating properties. To obtain this, Zhang et al. prepared MgO nanoparticle-decorated CFs and blended them into Nylon 66 [9]. The addition of CF lead to enhancements in thermal conductivity. However, MgO nanoparticles helped in quenching the electrical conductivity. Osouli-Bostanabad et al. fabricated nano-magnetite coated CFs that not only had enhanced strength, but also acted as an electromagnetic (EM) shield [10]. Briefly, they manufactured a double-layered composite material where the first layer provided good mechanical strength and the second layer aided in absorbing stray EM radiations. Thermal protection systems (TPS) are extensively incorporated in spacecraft where ablative polymer composites are majorly used to manufacture the system components. Naderi et al. prepared a nano Zirconia modified phenolic resin/CF composite that showed superior ablative properties in addition to greater mechanical strength [11]. Due to its high carbon content, outstanding thermal stability and reduced thermal expansion, CFs are usually employed as ablative materials. With the addition of nano Zirconia as a filler material, the thermal, insulative and ablative properties of the composite material were significantly enhanced.

CF reinforced polymers are used in various industries. However, due to certain restrictions of CF like smooth graphitic surface and diminished surface energy, they cannot form a good bond with the matrix. Thus, to enhance the interfacial adhesion between the fiber and the matrix, Jager et al. used halloysite nanotubes (HNT) and they observed significant improvement in interfacial shear strength (IFSS) with just 5% HNT [12]. Blugan et al. modified Alumina matrix with different concentrations of CNF to enhance the electrical conductivity of the matrix [13]. MWCNTs have long been used to modify and enhance the performance characteristics of the matrix material. However, they possess some

technical challenges like non-uniform dispersion throughout the matrix and weak bonding with the matrix. The use of CNF not only aids in eliminating these issues but also provides a more economical solution since they are cheaper than MWCNT. Li et al. performed tensile creep studies of CF/epoxy resin and MWCNT/CF/epoxy resin composites and found that the latter fared well in all the mechanical tests performed on them [14]. They proved that addition of MWCNT into CF composites can significantly boost their performance. Pervin et al. prepared nano composites using CNF and SC-15 epoxy [15]. Increased percentage of CNF resulted in higher values of modulus and mechanical strength parameters. With 4% addition of CNF, the modulus and strength of the composite material improved by 27% and 17%, respectively. Additionally, the composite also displayed better thermal response characteristics like enhanced glass transition temperature ($T_g$). Charles et al. modified CF–epoxy resin composites using a triblock copolymer of poly (styrene)-b-poly (butadiene)-b-poly(methylmethacrylate) [16]. The results showed that the initiation fracture toughness improved by 88% and there was a 6-degree Celsius rise in $T_g$. Furthermore, there was a 121% increment in shear strain with just an 8% decrement in shear strength. The superior performance of the triblock copolymer toughened composite was due to nano-structuring that took place inside the resin system and led to matrix cavitation, which resulted in easy dissipation of strain energy.

Figure 1 below shows the SEM image of electrochemically treated CF, apparatus used for carbonization of the nano sheets, schematic representation of the process used for fabrication of CF–MgO hybrid nanocomposites and stress–strain curves for SC-15 epoxy nanocomposite with different percentages of CNF.

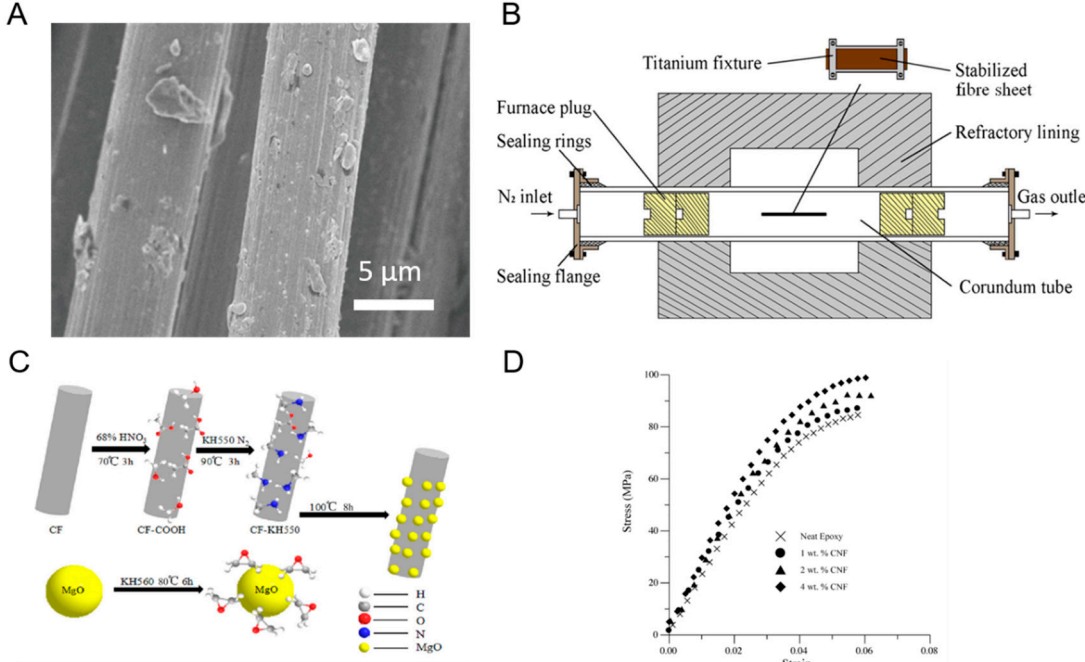

**Figure 1.** (**A**) SEM image of electrochemically treated Carbon Fiber. Reproduced with permission from [2]. (**B**) Schematic of the apparatus used for carbonization of nanosheets. Reproduced with permission from [6]. (**C**) Illustration of the process for generation of carbon fiber (CF)-MgO hybrid nanocomposite. Reproduced with permission from [9]. (**D**) Stress–strain curves for SC–15 epoxy nanocomposite for different weight percentages of carbon nano fiber (CNF). Reproduced with permission from [15].

## 2. Carbon Fibers in Renewables

The renewable energy sector has evolved to a great extent with alternative energy sources like secondary rechargeable batteries, supercapacitors, solar cells and fuel cells. In this section we will

discuss hybrid CNF-based renewable devices that have enhanced the performance characteristics of these devices.

## 2.1. Rechargeable Batteries

Secondary rechargeable batteries can provide an excellent replacement for conventional sources of energy. Additionally, they can serve as a back-up to other sources in a distributed generation system. Here, we briefly take a look at the various hybrid CNF-based vanadium redox flow [17,18], sodium-ion [19], lithium oxygen [20], lithium-ion [21–25] and zinc–carbon26 batteries that have been designed by researchers.

Vanadium flow batteries (VFB) provide many benefits like high efficiency, durability, being environment-friendly, and they also avoid the common problem of ion crossover. However, the commercialization process of these batteries is a challenging aspect because of technical barriers associated with the electrode, separator and electrolyte. A bipolar plate is an important aspect of these batteries since it gives a path for conduction of electrons and it also separates cells. Nam et al. fabricated a CF/fluoroelastomer composite bipolar plate [18]. Compared to earlier versions, the present bipolar plate manufactured by them gave high electrical conductivity and chemical stability against oxidizing and acidic conditions. The fluoroelastomer enabled the composite to have a high volume that led to enhanced electrical conductivity. Mechanical tests performed on the composite plate revealed good strength with a Young's modulus of 48.5 GPa and a Poisson ratio of 0.34. Moreover, the composite showed good electrical performance characteristics where the energy efficiency was 80.4% at a current density of 1000 $A/m^2$. Sodium-ion batteries (SIB) have recently gained high popularity and they can perform similar to lithium-ion batteries (LIB) and sometimes even outperform them. Electrochemically, both Na and Li function similarly. However, due to high availability of Na, the cost of SIBs is much cheaper than LIBs. However, due to larger radius and smaller diffusion of Na ions as compared to Li ions, it becomes difficult for Na ions to enter graphite, which is the common electrode used for LIB. Due to the unique electrical and chemical properties of Transition Metal Dichalcogenides (TMDCs) like $MoS_2$, they have been intensively incorporated as electrode materials. However, due to large volume change during sodiation/desodiation process, $MoS_2$ suffers from weak intrinsic conductivity and fast capacity reduction. To combat these difficulties, Chen et al. used $MoS_2$/electrospun CNF as a new composite material for SIBs [19]. The use of CNF restricts the volume change of $MoS_2$ and also enhances its conductivity. The fabricated electrode had good interfacial contact between the two materials used in the composite which led to improved cycling stability and better rate performances. The modified SIB had a high charge capacity of 380 mA h $g^{-1}$ after 50 charge–discharge cycles. Additionally, after 500 cycles, it could still provide a charge capacity of 198 mA h $g^{-1}$ at a high current density of 1 A $g^{-1}$.

Figure 2 below shows the fabrication procedure and SEM image of the CS/GF electrode used in VFB, and generation method as well as sodium-storage mechanism of the $MoS_2$@CNF electrode incorporated in SIB.

Non-aqueous lithium–oxygen batteries (LOB) have many advantages over LIB like the usage of oxygen from the atmosphere instead of having heavy reactants that are normally used in LIB. However, due to the slow oxygen reduction reaction (ORR) and oxygen evolution reaction (OER), LOB have low cycling performance, large over-potential, poor round-trip efficiency and restricted rate capability. Since both ORR and OER occur at the cathode, it is essential to have high conductive electrodes. Cao et al. fabricated Co loaded CNF as a free-standing cathode for $LiO_2$ batteries [20]. The addition of Co in proper amounts can result in a continuous and porous CNF that enhances its surface area and facilitates electron transfer and oxygen diffusion. The fabricated Co/CNF films have more active sites that encourage reactant diffusion and can also store vast amounts of $Li_2O$. Due to good catalytic performance of Co, the Co/CNF electrode enhances the capacity, rate capability and cyclic stability of LOB. 7.4 wt% of Co can give maximum benefits. However, overloading the CNF with 11.1 wt% of Co can result in disorientation of the film structure and can make it fragile. The LOB based on Co/CNF

electrode had a charge capacity of 4583 mA h g$^{-1}$ at a current density of 100 mA g$^{-1}$. Additionally, after 40 charge–discharge cycles, it still displayed a charge capacity of 500 mA h g$^{-1}$ at 100 mA g$^{-1}$.

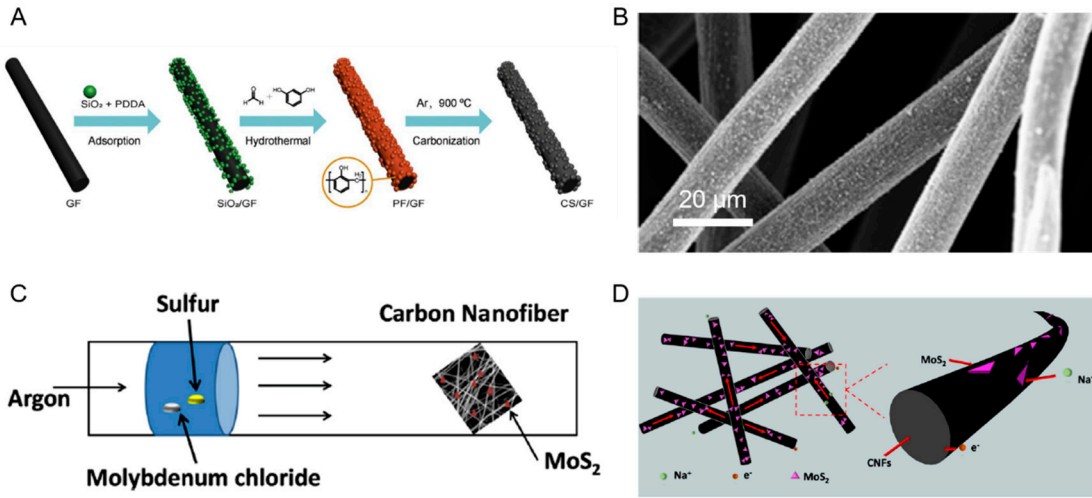

**Figure 2.** (**A**) Fabrication method of Carbon Sphere(CS)/glass fabric (GF) electrode. (**B**) SEM image of CS/GF electrode. Reproduced with permission from [17]. (**C**) Schematic depiction of the method of Figure 2. CNF composite. (**D**) Pictorial representation of sodium storage mechanism in MoS2@CNFs. Reproduced with permission from [19].

Recently, the demand for flexible battery systems has increased dramatically because of the introduction of smart devices and wearable electronics in the market. Hence, it is essential to develop flexible electrodes. Most LIBs make use of graphite as the electrode material. However, because of its low specific capacity of 372 mA h g$^{-1}$, its use is limited. Due to the advantage of high theoretical specific capacity of Si, Si-based alloys are being researched as the new electrode material for LIBs. However, during the charge–discharge cycles, the entry of Lithium ions into the Si electrode causes it to expand by almost 400%. This results in the loss of ohmic contact between the Carbon conductor and Si that further causes degraded electrical performance. To meet this drawback, Dirican et al. fabricated a free-standing and flexible Silicon/Silica/Carbon (Si/SiO$_2$/C) nano fiber composite as an anode material for LIBs [21]. The addition of SiO$_2$ helped in controlling the expansion of Si during multiple charge–discharge cycles. The generated nano fiber composite was further coated with nanoscale carbon by chemical vapor deposition (CVD). This further helped in maintaining the Si nanoparticles within the composite. The CVD carbon-coated fiber composite showed high capacity retention and coulombic efficiency of 86.7% and 96.7%, respectively, at the 50th charge cycle.

In another fiber-based battery system for flexible electronic applications, Yu et al. designed CF electrodes for Zinc–Carbon (Zn–C) batteries [26]. Zn–C batteries have grown enormously since their inception into the market because of their low cost due to frugal raw materials, low internal resistance and high energy density. These batteries utilize Zn as the cathode and MnO$_2$-graphite powder as the anode. The researchers modified the composition of the electrode materials and coated the conventionally used Zn and MnO$_2$-graphite powder on CF. The modified battery had an open circuit voltage of 1.5 V. Additionally, at the discharge density of 70 mA g$^{-1}$, the battery had a discharge capacity of 158 mA h g$^{-1}$. Moreover, the fiber added flexibility to the battery. Upon changing the bending radius from 3 cm to 0.7 cm, the battery did not show any degradation in its performance. Furthermore, when the fiber length expanded to 8 cm from 2 cm, the discharge capacity remained intact.

## 2.2. Supercapacitors

Similar to batteries, supercapacitors provide higher power but have lower specific energy. Owing to the conductive nature of CF, they have been widely employed in making modern-day supercapacitors that have enhanced output performance characteristics [27–36].

$MnO_2$ electrode materials provide many benefits due to their low cost, environmental compatibility and abundance. Chi et al. fabricated Boron-doped $MnO_2$/CF composites intended to be used as electrodes in supercapacitors [27]. The addition of Boron enhanced the growth rate of $MnO_2$ crystals. Doping improves the specific capacitance and cyclic stability of supercapacitors. Thus, the $MnO_2$/CF composite electrode supported the electrochemical reactions and enhanced the surface charge storage and rate capabilities of the supercapacitor. Even after 1000 charge–discharge cycles, the supercapacitor retained 80% of its initial capacitance. The composite fiber had a worm-like structure that led to an increased specific capacitance of 364.8 F $g^{-1}$ and a surface charge density of 19.5 C $g^{-1}$.

Polymeric capacitors that are deposited on carbon materials show enhanced performance. Davoglio et al. prepared thin films of Polypyrrole (PPy) and poly-2,5-dimercapto-1,3,4-thiadiazole (poly (DMcT)) coated on CF cloth [28]. The bilayer composite was prepared by coating PPy on poly (DMcT)-functionalized CF. The addition of PPy helped in preserving poly (DMcT) and hence bettered its charge-storage capabilities after 1000 charge–discharge cycles. The composite was lightweight and had a high surface area. The CF/poly (DMcT)/PPy composite had a high specific capacity value of 320 mA h $g^{-1}$. The CF/PPy and CF/poly (DMcT)/PPy composites had specific capacitance values of $460 \pm 50$ and $1130 \pm 100$ F$g^{-1}$, respectively.

Xie et al. generated coaxial micro fibers (CMF) that were comprised of Ni and CNT coated on CF [29]. The purpose of functionalizing CF was to increase its specific surface area. To increase the energy storage capacity, they coated Ni on CF and then added CNT. The fabricated CMF had higher surface area, electrical conductivity and capacitance. Additionally, the CMF had good tensile strength to be used as electrode material for flexible electronic applications. The CF–Ni–CNT composite had 1400 and 100 times higher specific surface area (SSA) and capacitance, respectively, as compared to bare CF.

Bare Carbon fiber papers (CFP) have been widely used in batteries, supercapacitors and fuel cells. However, because of weak ionic conductivity and hydrophobic surface, they cannot store a high amount of charge. Suktha et al. designed functionalized CFP in order to overcome these shortcomings [30]. They functionalized it mainly with carboxyl (-COOH) and hydroxyl (-OH) groups. The functionalized CFP (*f*-CFP) exhibited good performance characteristics. It displayed areal, volumetric and specific energies of 49 μW h $cm^{-2}$, 1960 mW h $L^{-1}$ and 5.2 W h $kg^{-1}$ and powers of 3 mW $cm^{-2}$, 120 W $L^{-1}$ and 326.2 W $kg^{-1}$, respectively.

Figure 3 below shows the X-ray diffraction (XRD) patterns for CF, undoped electrode and Boron-doped electrode, the SEM images of CF/poly(DMcT)/PPy composite and CNTs grown on CFs and the Fourier Transform Infrared spectroscopy (FTIR) spectra of bare and functionalized CFP.

Yin et al. fabricated a flexible and conductive thin film made of Polydimethylsiloxane (PDMS), Ag nanowires (AgNWs) and CFs [31]. Along with improving the electrical conductivity, CF helped in strengthening the composite by helping it to resist any mechanical deformations. Moreover, the addition of AgNWs assisted in increasing the surface area of CF and in reducing the contact resistance between the adjacent CFs. The composite film had a low sheet resistance of 0.99 Ω $m^{-2}$. Additionally, even after 275 consecutive cycles of bending and releasing processes, the sheet resistance decreased by 3%. Interestingly, the increased addition of CF enhanced the load bearing capability of the composite. Increasing the amount of CF from 50 mg to 200 mg resulted in better stress–strain curves. The resistance of the composite with a higher content of CF was also lower. The resistance decreased from 50.1 Ω for 50 mg CF to 19.1 Ω for 100 mg CF and 9.6 Ω for 200 mg CF. In another application of developing flexible electrodes for supercapacitor application, Ma et al. fabricated a unique composite by depositing Nickel Hexacyanoferrate nanocubes (NiHCF-NCs) on flexible CFs [32]. The generated electrode had a high capacitance of 476 F $g^{-1}$ at 0.2 A $g^{-1}$. Additionally, the electrode was able to

retain 92.5% of its capacitance even after 8000 charge/discharge cycles. Nanostructured NiHCF helps in overcoming the shortcomings of volume expansion and extreme agglomeration that are usually faced by bulk-sized NiHCF.

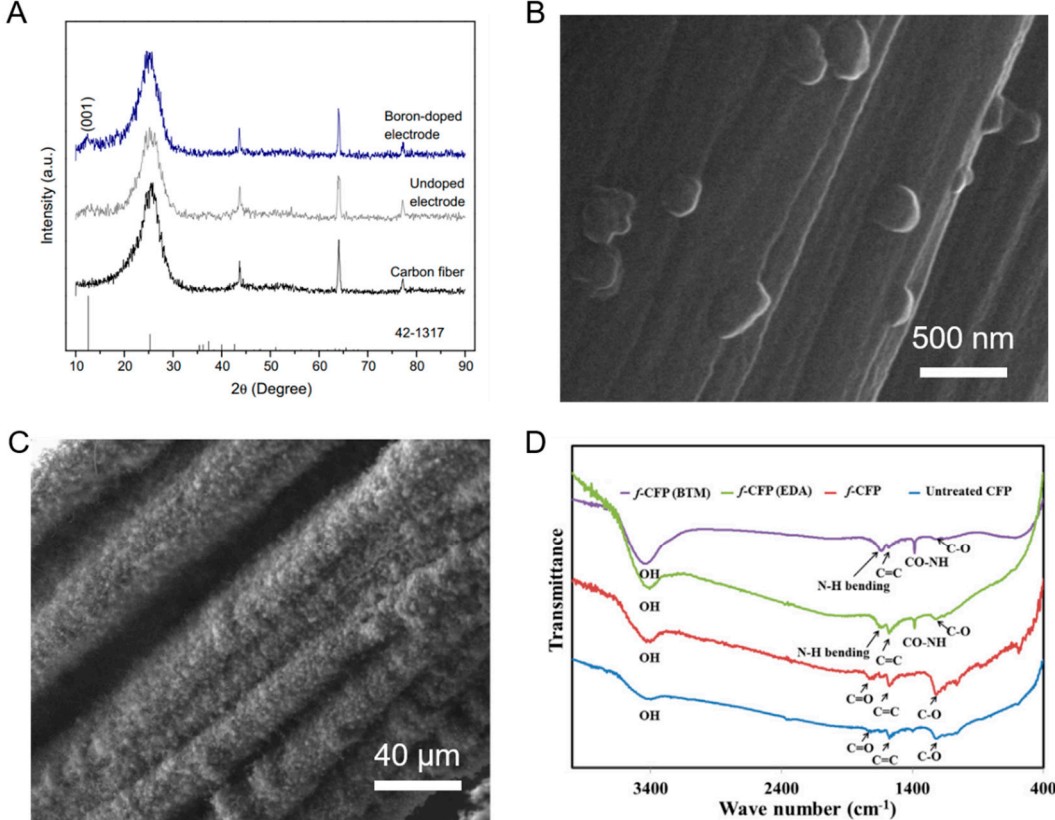

**Figure 3.** (**A**) XRD patterns for CF, undoped electrode and Boron-doped electrode. Reproduced with permission from [27]. (**B**) SEM image of CF/poly-2,5-dimercapto-1,3,4-thiadiazole (DMcT)/Polypyrrole (PPy) composite. Reproduced with permission from [28]. (**C**) SEM image of carbon nanotubes CNTs grown on CFs. Reproduced with permission from [29]. (**D**) FTIR spectra of bare and functionalized carbon fiber paper CFP. Reproduced with permission from [30].

The poor electrical conductivity of $MnO_2$ restricts its usage in supercapacitors as the rate capabilities are significantly lowered because of fall of specific capacitance. Thus, Zhao et al. fabricated ZnO@Au@$MnO_2$ nanosheets on CF paper [34]. Due to this hierarchical structure, the electrical conductivity and specific capacitance is enhanced as there is large electric contact that increases ion and electron transport rate and also shortens the ion diffusion path. The specific capacitance of the composite material was 654 F $g^{-1}$, calculated using cyclic voltammetry (CV), and 478 F $g^{-1}$ at a current density of 2.6 A $g^{-1}$. Furthermore, it could retain 80% of its capacitance after 2500 charge–discharge cycles. The presence of CF paper makes the composite light weight and thus makes it a promising candidate for applications in smart electronics. Dong et al. prepared mesoporous graphitic carbon fibers that had large surface areas and high pore volumes of 870–1790 $m^2g^{-1}$ and 0.729–1.308 $cm^{-3}g^{-1}$ [35]. The prepared fibers had a high specific capacitance of 303 F$g^{-1}$ at 0.7 A$g^{-1}$ that could be retained well after several charge/discharge cycles. The enhanced performance was due to the morphology of the fiber, better porous structure and degree of graphitization.

## 2.3. Solar Cells

Harnessing solar energy has been an enormous blessing as it has reduced the burden on conventional generators and has also helped in lessening the negative impacts of non-renewable

sources of energy. Here, we look into some structural configurations of dye-sensitized solar cells (DSSCs) that utilize hybrid CF materials as electrodes [37–46].

Fibers made of Carbon nanomaterials have high-performance characteristics. Fang et al. fabricated a unique core-sheath carbon nanostructure fiber [37]. The structure comprised of a CNT core that provided high tensile strength and electrical conductivity and gold nanoribbons (GNRs) that gave high electrocatalytic activity. Similar to the bare CNT fiber, the core-sheath type composite fiber had electrical conductivities and tensile strengths of $10^2$–$10^3$ S cm$^{-1}$ and $10^2$–$10^3$ MPa, respectively. However, as an added advantage, the sheath helped in achieving high catalytic activity as the atomic edges were exposed on the surface. As compared to other structures (CNT/Graphene Oxide (GO) fiber, CNT fiber, GNR fiber), CNT/GNR displayed the best output characteristics: Open Circuit voltage, $V_{oc}$ of 0.7 V, current density, $J_{sc}$ of 12.07 mA cm$^{-2}$, fill factor (FF) of 60.95% and an efficiency, $\eta$ of 5.16%.

Veerappan et al. replaced conventional Pt electrodes with CNFs as the counter electrode (CE) for DSSCs [38]. Because of the nano structured morphology, the CNFs-CE had a faster $I_3^-$ reduction rate and low charge transfer resistance $R_{CT}$ of 0.5 $\Omega$ cm$^{-2}$ as compared to Pt. Because of specific characteristics like terminating graphitic layers on the fiber surface, large defects on edge planes and big pore diameters as well as rough and large surface areas, CNFs have faster electron transfer mechanics that lead to enhanced photovoltaic output characteristics.

The improved performance of CEs in DSSCs requires low charge-transfer resistance and high electrocatalytic activity of the active material used as the CE. Due to the high cost of Pt, other transition metal structures are being researched. Yousef et al. fabricated NiCu nanoparticles (NP) that were coated with CNF and used them as the CE for DSSCs [39]. CNF provided good shielding against corrosion and also helped in enhancing electrical conductivity and adsorption capacity. The DSSC based on the Cu/Ni CNF composite CE had good photovoltaic output characteristics: $V_{oc}$ of 0.7 V, $J_{sc}$ of 7.67 mA cm$^{-2}$, FF of 65% and $\eta$ of 3.5%. In another Pt-free application, Yousef et al. prepared Co-TiC NPs embedded on CNF [40]. The composite was used as a CE in DSSCs and fuel cells (FCs). The photovoltaic output characteristics of the DSSC were $V_{oc}$ of 0.758 V, $J_{sc}$ of 9.98 mA cm$^{-2}$, FF of 50.7% and $\eta$ of 3.87%. The researchers attributed the enhanced electrocatalytic activity of the CE to the synergetic effects of its individual components.

Figure 4 below displays the SEM image of the core-sheath nanostructured fiber, the photovoltaic output characteristics for antler Carbon nanofiber (CNF–LSA) and Pt CE-based DSSCs and the Voltage-Current (V-I) performance for NiCu–CNF composite CE-based DSSC and Co-TiC CNF CE-based DSSC.

Chen et al. used a composite of PtNPs and vapor grown carbon fibers (VGCFs) coated on Fluorine-doped Tin Oxide (FTO) glass as a CE for DSSC [43]. Compared to CNT, VGCFs have poor mechanical characteristics. However, VGCFs have more structural defects due to which they have more active sites for electrocatalytic reactions. In order to have higher efficiency at a reduced cost, the researchers focused on preparing PtNPs/VGCF composites. The PtNPs increased the thermal stability of VGCFs and they were uniformly distributed over VGCFs, which aided in increasing the surface area that facilitated the redox reactions taking place inside the films. The hybrid PtNP/VGCF CE with a weight ratio of 3:7 for PtNPs to VGCFs, had a higher photovoltaic conversion efficiency of 7.77% as compared to 7.31% and 3.97% for the conventional Pt CE and bare VGCF CE.

To produce a low-cost alternative to conventional Pt-based electrodes, Chen et al. used transition metal compounds that had good catalytic activity and electronic structures that resembled that of Pt [44]. They fabricated $CoNi_2S_4$ nanoribbons on CFs as a CE for fiber-shaped DSSCs (FDSSCs). The CE made up of the $CoNi_2S_4$/CF composite material had an efficiency of 7.03%. The researchers also fabricated a different versions of the composite where they used $CoNi_2S_4$ nanorods on CFs. However, it displayed a low photovoltaic conversion efficiency of 4.10%. Additionally, while comparing the I-V curves of bare CF, $CoNi_2S_4$ nanorod-CF, Pt wire and $CoNi_2S_4$ nanoribbon-CF, the nanoribbon morphology gave the highest current density. The photovoltaic output characteristics of the DSSC made up of $CoNi_2S_4$ nanorod-CF were $V_{oc}$ of 0.68 V, $J_{sc}$ of 15.3 mA cm$^{-2}$ and FF of 67.7%.

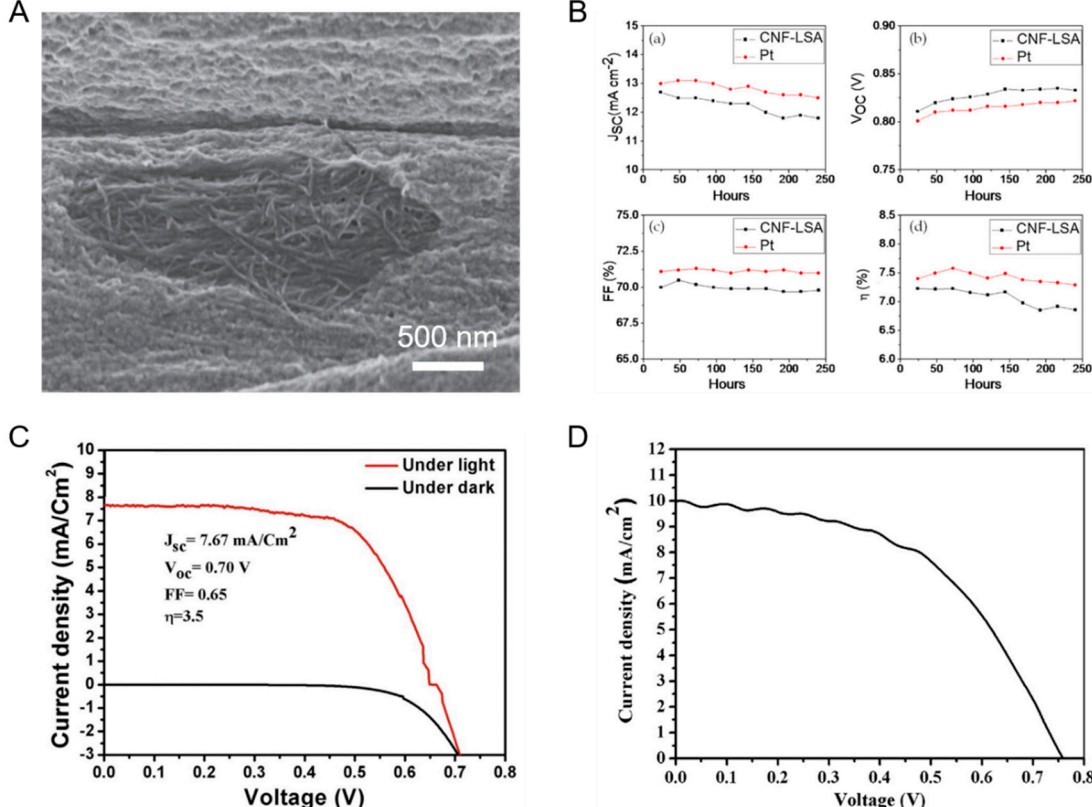

**Figure 4.** (**A**) SEM image of the core-sheath nanostructured fiber. Reproduced with permission from [37]. (**B**) Photovoltaic output characteristics ((a) short-circuit current density, $J_{SC}$, (b) open-circuit voltage, $V_{OC}$, (c) fill-factor, FF, (d) energy conversion efficiency, µ) for CNF-LSA and Pt counter electrode (CE)-based dye-sensitized solar cells (DSSC)s. Reproduced with permission from [38]. (**C**) V-I performance and cyclic voltammetery curves for NiCu-CNF composite CE-based DSSC. Reproduced with permission from [39]. (**D**) V-I performance curve of the Co-TiC CNF CE-based DSSC. Reproduced with permission from [40].

Guo et al. fabricated $TiO_2$ nanorod (NR) arrays grown on CF as photoanode for DSSC [45]. The NR-based solar cell had $V_{oc}$ of 0.63 V, $J_{sc}$ of 2.57 mA cm$^{-2}$, FF of 47% and $\eta$ of 0.76%. The CF-based DSSC was tube-shaped and it could capture light from all directions. Additionally, it showed high electrical conductivity, anti-corrosive property towards $I_2$ and high reactivity for triiodide reduction. Combining all these with the economical price of carbon materials, the CF-based solar cell could be a good alternative to Pt electrodes. In another hybrid application, Guo et al. fabricated Pt/CF composites for a CE that could be used for redox reactions of $Co^{3+}/Co^{2+}$, $T_2/T^-$ and $I_3^-/I^-$ [46]. With a low 1 wt% loading of Pt, the best output characteristics were obtained. With just bare CF, the output characteristics were $V_{oc}$ of 0.846 V, $J_{sc}$ of 13.49 mA cm$^{-2}$, FF of 56% and $\eta$ of 6.39%. With 1 wt% of Pt, the output characteristics of the composite were $J_{sc}$ of 15.52 mA cm$^{-2}$, FF of 68% and $\eta$ of 8.97%. The enhanced features of the DSSC could be attributed to the high catalytic activity of the Pt/CF composite towards the redox couples.

Table 1 below summarizes the output characteristics of the CF-based DSSCs discussed above.

**Table 1.** Summary of output characteristics of CF-based DSSCs.

| S.No. | Active Materials | Output Performance | Reference |
|:---:|:---:|:---:|:---:|
| 1. | CNT/GNR fiber | $V_{oc}$ = 0.846 V<br>$J_{sc}$ = 13.49 mA cm$^{-2}$<br>FF = 56%<br>$\eta$ = 6.39% | [37] |
| 2. | CNF-LSA | $V_{oc}$ = 0.779 V<br>$J_{sc}$ = 12.6 mA cm$^{-2}$<br>FF = 55.2%<br>$\eta$ = 5.4% | [38] |
| 3. | CuNi NPs-CNF | $V_{oc}$ = 0.7 V<br>$J_{sc}$ = 7.67 mA cm$^{-2}$<br>FF = 65%<br>$\eta$ = 3.5% | [39] |
| 4. | Co-TiC NPs-CNF | $V_{oc}$ = 0.758 V<br>$J_{sc}$ = 9.98 mA cm$^{-2}$<br>FF = 50.7%<br>$\eta$ = 3.87% | [40] |
| 5. | Hollow activated CNF | $V_{oc}$ = 0.73 V<br>$J_{sc}$ = 14.5 mA cm$^{-2}$<br>FF = 62%<br>$\eta$ = 6.58% | [41] |
| 6. | NiCo$_2$S$_4$-CF | $V_{oc}$ = 0.63 V<br>$J_{sc}$ = 17.78 mA cm$^{-2}$<br>FF = 56%<br>$\eta$ = 6.31% | [42] |
| 7. | PtNPs/VGCF | $V_{oc}$ = 0.55 V<br>$J_{sc}$ = 15.47 mA cm$^{-2}$<br>FF = 45%<br>$\eta$ = 3.79% | [43] |
| 8. | CoNi$_2$S$_4$ nanoribbon-CF | $V_{oc}$ = 0.68 V<br>$J_{sc}$ = 15.3 mA cm$^{-2}$<br>FF = 67.7%<br>$\eta$ = 7.03% | [44] |
| 9. | TiO$_2$ NR-CF | $V_{oc}$ = 0.63 V<br>$J_{sc}$ = 2.57 mA cm$^{-2}$<br>FF = 47%<br>$\eta$ = 0.76% | [45] |
| 10. | Pt/CF | $J_{sc}$ = 15.52 mA cm$^{-2}$<br>FF = 68%<br>$\eta$ = 8.97% | [46] |

*2.4. Fuel Cells*

In addition to the above-mentioned energy storage devices, fuel cells are also equally gaining popularity because of their compact size, easy decentralization and low maintenance requirements. Here, we summarize some recently developed hybrid CF-based fuel cells that have been designed to give enhanced performance.

In a unique application, Li et al. developed a soil microbial fuel cell (MFC) by mixing CF with petroleum hydrocarbon contaminated soil [47]. The addition of CF helped the anode in collecting more electrons and thus the maximum current and power density and accumulated charge output was enhanced 10, 22 and 16 times as compared to fuel cell made without the hybrid material. Moreover, the internal resistance of the cell reduced by 58% which lead to improvements in efficiency.

The researchers hence found that the use of conductive CF was beneficial for bioelectricity recovery from soil.

Gas diffusion layers (GDL) in proton exchange membrane fuel cells (PEMFCs) are important for maintaining $H_2$/air system performance in regions of high current density. It performs crucial tasks of distributing reactants to active sites, managing water supply and enhancing electrical contact between the electrode and bipolar plates. Kannan et al. prepared a micro-porous GDL with the aid of CNF and carbon nano-chain Pureblack [48]. The researchers found that addition of CNF enhanced the mechanical properties of the GDL. The fuel cell made with CNF-based GDL gave a high power density of 0.55 $Wm^{-2}$.

Okada et al. used CNF interlayer in a direct methanol fuel cell at the interface of carbon paper and a PtRu NP catalyst layer [49]. The dense and crackless CNF layer reduced catalyst loss and led to increased active reaction sites on the anode. The CNF layer helped in enhancing the electrical conductivity of the fuel cell and also enhanced the power density.

Lim et al. developed a CF/Poly Ether–ether Ketone (PEEK) composite bipolar plate for a high temperature PEMFC [50]. The composite enhanced the electrical conductivity and mechanical strength of the fuel cell. Furthermore, environmental durability tests confirmed the sustainability of the device.

Shu et al. prepared GDL made up of CF felt and Polytetrafluoroethylene (PTFE) for Mg-air fuel cells [51]. The prepared GDL showed enhanced mechanical properties and improved electrical conductivity along with water-repellent properties and high gas permeability as compared to Mg-air fuel cells based on a conventional carbon powder-based cathode.

CNF and activated CNF (ACNF) have been used as the cathode catalyst in MFCs [52]. The ACNFs enhance the catalytic activity sites because of their large surface area. Additionally, they are much more economical as compared to standard Pt cathodes. With enhanced physical properties, the CNFs provide great help in increasing the device performance.

Abdelkareem et al. used Ni–Cd CNFs as a catalyst for urea fuel cells [53]. The low commercialization of these cells is due to low catalytic activity of the anode. With the composite catalyst developed by the researchers, they were able to increase the number of active sites for urea oxidation and hence improve the device's electrical and mechanical output characteristics.

Hence, CF and CNF-based composites have been applied as electrode materials in renewable energy devices like batteries, supercapacitors, solar cells and fuel cells. With just minute additions of CF/CNF, the electrical, thermal and mechanical properties of the base material can be greatly enhanced. Due to features like high surface area, the number of catalytically active sites and the electron flow rate is improved to a high extent, leading to enhanced efficiency and power output of these devices.

## 3. Sensing Using Carbon Fibers

Several hybrid sensing schemes have recently been developed by researchers that incorporate CF. The inclusion of CF enhances the sensing properties of the composite material. In this section, we describe in depth some of the glucose [54], cortisol [55], neurotransmitter [56], Bisphenol-A [57], Ethanol [58], NO and CO [59], Hydrogen [60], Uracil and 5-Fluorouracil [61], Hydrogen Peroxide [62], DNA [63] and strain [64–67] and chemoresistive [68] gas sensors that are based on hybrid CF materials.

Weina et al. fabricated a glucose sensor by coating $\beta$-$MnO_2$ micro/nanorod arrays on a CF fabric [54]. $\beta$-$MnO_2$ was chosen as the active material for the bio-sensor because of its favorable physical and electrochemical properties. The coating of $\beta$-$MnO_2$ on CF further enhanced its desirable properties. CF led to higher porous channels for electrolyte diffusion and also facilitated ion transport. The sensitivity of the glucose sensor was 1650.6 $\mu A$ $mM^{-1}$ $cm^{-2}$. The detection limit was 1.9 $\mu M$ and the linear range was 0.01–4.5 mM. The composite sensor showed high sensitivity, selectivity and stability. The improved performance of the CF-based sensor was due to an increased number of electrochemically active sites, appropriate utilization of $\beta$-$MnO_2$, faster mass transport and direct contact with the current collector.

Sekar et al. used conductive carbon yarn (CCY) to manufacture a wearable sensor for measurement of sweat cortisol [55]. Due to the crystalline nature of CCY, the fiber showed high strength-to-volume ratio. Briefly, they functionalized CCY with $\alpha$-Fe$_2$O$_3$ in order to immobilize antibodies specific to cortisol. The cortisol sensor had a detection limit of 0.005 fg mL$^{-1}$ and had a linear range from 1 fg to 1 μg with a high correlation factor of R$^2$ = 0.998. Additionally, the sensor results were repeatable, reliable and reproducible. Moreover, it showed good selectivity towards cortisol when compared to other analogous compounds like cortisone, progesterone and cholesterol. Moreover, it had a short response time of 120 s.

Fiber-reinforced polymer composites often undergo complex and multi-phase failures. The initiation of the damage process begins with the appearance of microcracks. Incorporation of carbon materials in the composites can help in self-diagnosis of cracks. In this regard, Gallo et al. modelled and characterized self-sensing of microcracks of CNT and CF-based composites [64] There was a direct correlation between electrical conductivity of the composite and applied mechanical stress. It was observed that increased stress resulted in lower electric conductivity. The self-sensing capability of the CF-based polymer composites can be a huge benefit for timely detection of cracks, which if undetected can pose a significant threat to civil and mechanical structures and can adversely affect and put several lives in danger.

Figure 5 below shows a schematic depiction of the fabrication process of a $\beta$-MnO$_2$ nanorod array on CF fabric and the XRD pattern of $\beta$-MnO$_2$ nanorod array for the glucose sensor, an illustration of the hydrothermal process of generating Fe$_2$O$_3$/CCY hybrid electrode and the Field Emission SEM (FESEM) image of the Fe$_2$O$_3$/CCY electrode for the cortisol sensor and a schematic of the transverse crack insertion between the electrode gage length and electrical conductivity summary for different carbon-based composite polymers for the self-sensing CF-based micro-crack detection sensor.

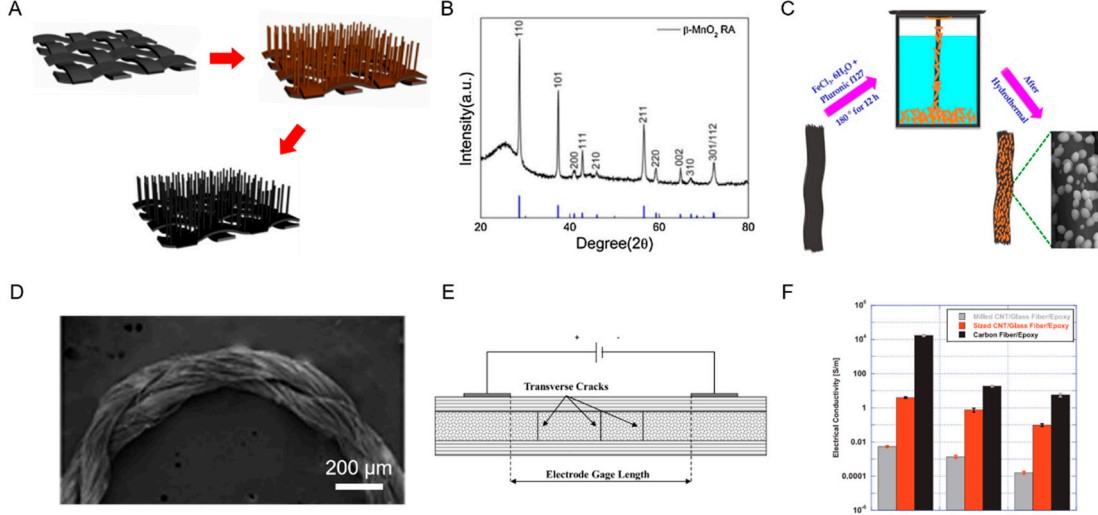

**Figure 5.** (**A**) Schematic depiction of fabrication of $\beta$-MnO$_2$ nanorod array on CF fabric. (**B**) XRD pattern of $\beta$-MnO$_2$ nanorod array. Reproduced with permission from [54]. (**C**) Illustration of the hydrothermal process of generating Fe$_2$O$_3$/conductive carbon yarn (CCY) hybrid electrode. (**D**) the Field Emission SEM (FESEM) image of the Fe$_2$O$_3$/CCY electrode. Reproduced with permission from [55]. (**E**) Schematic of transverse crack insertion between the electrode gage length. (**F**) Electrical conductivity summary for different carbon-based composite polymers. Reproduced with permission from [64].

Using electrospinning and heat treatment, Im et al. fabricated a CF composite structure for sensing NO and CO [59]. The generated CF was functionalized to improve the number of gas adsorption sites. Additionally, Carbon Black (CB) derivates were added to enhance the electrical conductivity. Due to the functionalization and addition of CB, the sensitivity was improved five times. The CB additives formed an electrically conductive network inside the fiber structure and the porous structure on the

outside of the fiber which resulted in higher gas adsorption. Thus, all these features led to higher sensitivity and selectivity of the sensor towards NO and CO gases.

Ou et al. prepared Pd–Ni nanofilms and NPs of various sizes and electrodeposited them on CF for hydrogen gas sensing [60]. In the concentration ranges of 0–2.8% and 3.6–6% $H_2$ gas, the response of the sensor increases with increasing concentration. However, for concentrations between 2.8–3.6% $H_2$ gas, the response decreases for the nanofilm sensor because of rearrangements in the molecular structure. Alternatively, the NP-based sensor displayed consistent results for the entire 0–6% $H_2$ gas concentration range. Increasing the concentration of $H_2$ gas results in higher conductivity. This was attributed to the diffusion of hydrogen gas inside the sensor that causes the NPs to expand, resulting in higher contact points between the particles.

Prasad et al. fabricated a silica-molecularly imprinted polymer (MIP) composite fiber using carboxylated MWCNTs for sensing ultra-trace levels of Uracil (Ura) and 5-Fluorouracil (5-FU) [61]. The carboxylated MWCNT enhanced the electrochemical reaction because of its edged plane-like surface area. The use of composite carbon-based electrodes can offer many benefits like greater binding sites and electrochemical activity as compared to traditional electrodes.

The generated sensor was cost-effective, disposable and reliable and had detection limits of 1.3 ng mL$^{-1}$ and 0.56 ng mL$^{-1}$ for Uracil and 5-Fluorouracil, respectively.

Figure 6 below shows the resistive response and FESEM image of the CF-based electrode for NO gas sensing, the SEM image and response curve of the CF-composite Pd–Ni alloy NP and nanofilm hydrogen sensor and the schematic and SEM image of the Silica–MWCNT–MIP composite fiber for Ura and 5-FU sensing.

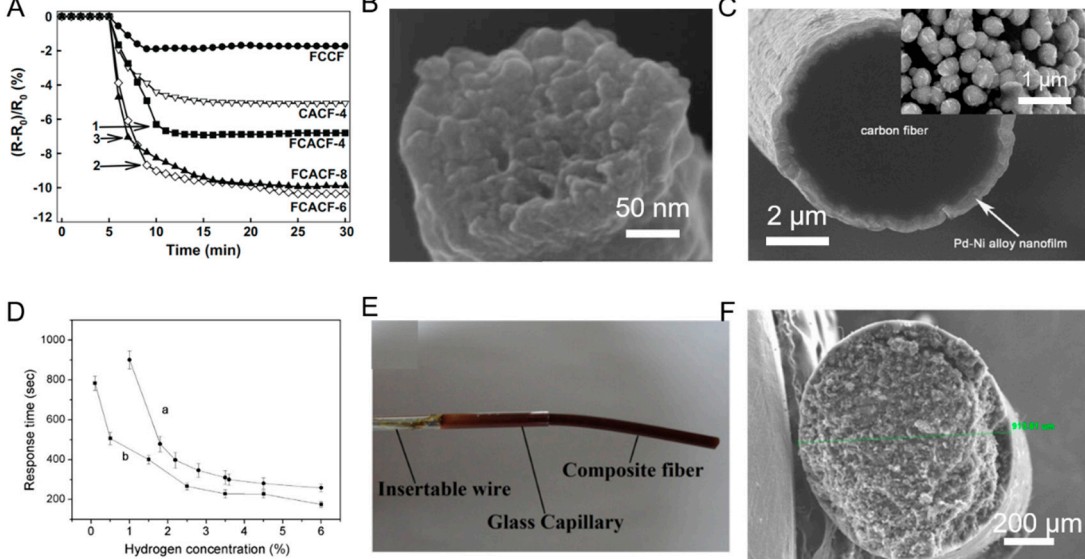

**Figure 6.** (**A**) Resistive response for NO gas sensing. (**B**) FESEM image of the CF-based electrode. Reproduced with permission from [59]. (**C**) SEM image of Pd–Ni alloy nanoparticle (NP) and nanofilm (insert). (**D**) Response curve of (a) nanofilm type hydrogen sensor and (b) CF-composite Pd–Ni NP. Reproduced with permission from [60]. (**E**) Schematic of Silica– Multi-walled Carbon Nanotube (MWCNT)– molecularly imprinted (MIP) composite fiber. (**F**) SEM image of MIP–5-Fluorouracil (5-FU) adduct. Reproduced with permission from [61].

Wu et al. fabricated a Pt NP doped carbon fiber ultramicroelectrode (Pt/CFUME) for an amperometric biosensor for detecting $H_2O_2$ [62] The response was measured in terms of output current. The response current was linearly related to $H_2O_2$ concentration in the range of 0.64 μM to 3.6 mM with a correlation factor of 0.9953 and the detection limit was 0.35 μM. The addition of Pt NPs

helped in faster electron transport that led to higher sensitivity. Additionally, the CF ultramicroelectrode increased the effective surface area leading to enhanced mass transport.

In another gas-sensing application, Calestani et al. prepared CFs that were functionalized by ZnO nanowires [68]. They generated a smart material that could be used as a piezoelectric strain as well as a chemoresistive gas sensor. Due to the crossing nature of the CF, the sensor can be modified into a non-invasive integrated array of structures having dimensions in microns and each array can be utilized for sensing a different signal. Hence, it is possible to achieve a wide number of sensing applications with just one single CF composite material.

As a biosensing application, Dogru et al. prepared a DNA biosensor using CF as a microelectrode and Methylene Blue (MB) as an indicator of hybridization [63]. The response was measured in milliamperes. The sensitivity of the biosensor was 0.19 mA $\mu M^{-1}$ and the linear range was from 0.08–1.6 $\mu M$ with a correlation factor of 0.995. Additionally, the detection limit was 0.12 $\mu M$.

Figure 7 below shows the SEM image of Pt/CFUME, the response curve for the $H_2O_2$ sensor, and a schematic and an SEM image of the two crossing CFs functionalized with ZnO NWs.

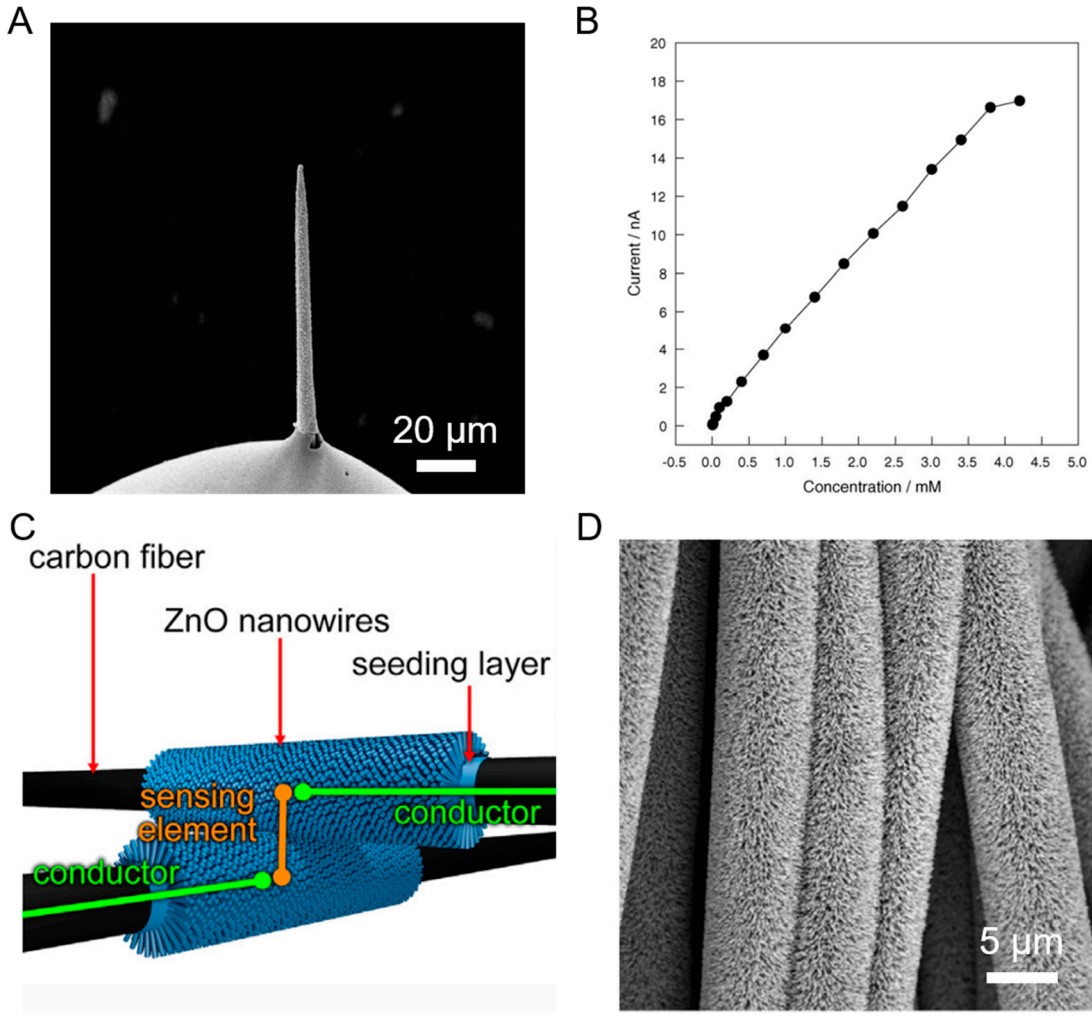

**Figure 7.** (**A**) SEM image of Pt NP doped carbon fiber ultramicroelectrode (Pt/CFUME). (**B**) Response curve for the $H_2O_2$ sensor. Reproduced with permission from [62]. (**C**) Schematic of the two crossing CFs functionalized with ZnO nanowires (NWs). (**D**) SEM image of CF functionalized with ZnO NWs. Reproduced with permission from [68].

Again, favorable thermal, mechanical and electrical features of CF and CNF can be highly beneficial for sensing applications. With more surface area, the electrical conductivity is enhanced

and even heat dissipation is better due to which the thermal limit and operating range of the sensor is improved. Moreover, with an increased number of sensing sites, the detection limit of the sensors can also be reduced.

## 4. Tissue Engineering Using Carbon Fibers

Tissue Engineering is a field of regenerative medicine and nanomaterials play an essential role as they impart sturdiness to various prosthetics and implants that are designed for the ailment of affected limbs and other human body parts. Carbon Fiber offers multiple advantages for repair of damaged cells and tissues. In this section, we briefly take a review of several composite CF-based materials that have been fabricated for the purpose of healing injured tissues.

Dentistry makes use of several glass fiber-based implants. However, CF can give several advantages over glass because of higher stiffness and strength, low abrasion and thermal expansion and chemical inertness towards most materials. Menini et al. examined several CF-based dental implants and performed destructive and non-destructive tests on them and found that they provided similar results to conventional metal-based implants [69]. Additionally, biological compatibility was tested using 3-(4,5-dimethylthiazol-2-yl)-2,5-diphenyltetrazolium bromide (MTT ) assay and they found that the cell vitality for the CF-based implant was 91.4%. Peterson fabricated a bisphenyl-polymer/carbon fiber reinforced composite and found that it could replace Titanium alloy bone implants quite effectively [70]. The composite could offer electrical conductivity properties that were similar to bones and were much higher than metallic implants. Radiation therapy for orthopedic patients having bone implants can have certain limitations like back scattering and beam attenuation due to the implants that can compromise and reduce the therapeutic effect of radiation. Laux et al. fabricated Carbon Fiber Polyether Ether Ketone (CF:PEEK) composite implants and found that they led to much lower attenuation for the radiation as compared to standard Titanium implants [71]. Rajzer et al. prepared CNFs by using PAN/hydroxyapatite (HAp) precursors and used them for bone tissue engineering [72]. The researchers found that upon insertion of the composite in simulated body fluid (SBF), the surface of the composite was covered with bone-like apatite. Huang et al. fabricated a CF-based platform for wound healing [73]. They performed in vivo analysis and found that fibroblast cells that had the three-dimensional (3D) CF scaffold could accelerate treatment of the wound. Furthermore, the fibroblast cells migrated to the wound location and increased the production of fibronectin and type I collagen that resulted in faster healing than other conventional treatment methods.

Figure 8 below shows the dental implant made of CF composite, a pictorial image of the Bisphenyl-polymer/CF implant and tissue and also the bone prepared with the CF/PEEK implant, and a schematic of the electrospinning apparatus for generating CNF.

Naskar et al. fabricated a biocompatible composite using CNF and nonmulberry silk fibroin and used them for regeneration of bones [74]. The compressive modulus of the composite is 46.54 MPa, which is quite a bit higher than the minimum required human trabecular bone modulus of 10 MPa. The composite showed minimum toxicity both in vitro and in vivo as there was minimum release of pro-inflammatory cytokines. Carbon fiber-reinforced polyetheretherketone (CFRPEEK) composites are ideal for implants. However, due to biological inertness and poor osteogenic properties they have limited applications. Xu et al. modified CFRPEEK composites by adding hydroxyapatite (HA) and produced nano-sized PEEK/CF/n-HA ternary composites [75]. The new composite demonstrated outstanding biocompatibility and also bonded well with the bones. In another application, Srivastava et al. developed tricalcium phosphate–polyvinyl alcohol doped CF reinforced polyester resin composites and tested them by implanting them in the artificially created bone defects in the bone marrow of rabbits [76]. The composite showed high compressive, tensile and bending strengths. They found the formation of cancellous bone around the implant region after 12–32 weeks of implantation. In an altogether different study, PROKIJ et al. used Magnetic Resonance Imaging (MRI) to evaluate its efficacy in monitoring the biocompatibility of surface functionalized CF implants [77]. They analyzed the interaction of CF implants with subcutaneous and muscular tissues of rabbits

and found that the MRI technique was quite beneficial in studying the biocompatibility of CF-based implants. Garcia-Ruiz et al. developed 3D structures using CFs and functionalized it with human mesenchymal stem cells (h-MSC) and found that the scaffold supported cell adhesion, proliferation and viability [78]. The 3D structure has the potential for not only repairing tendons and ligaments but also for regenerating cartilage and endochondral bone.

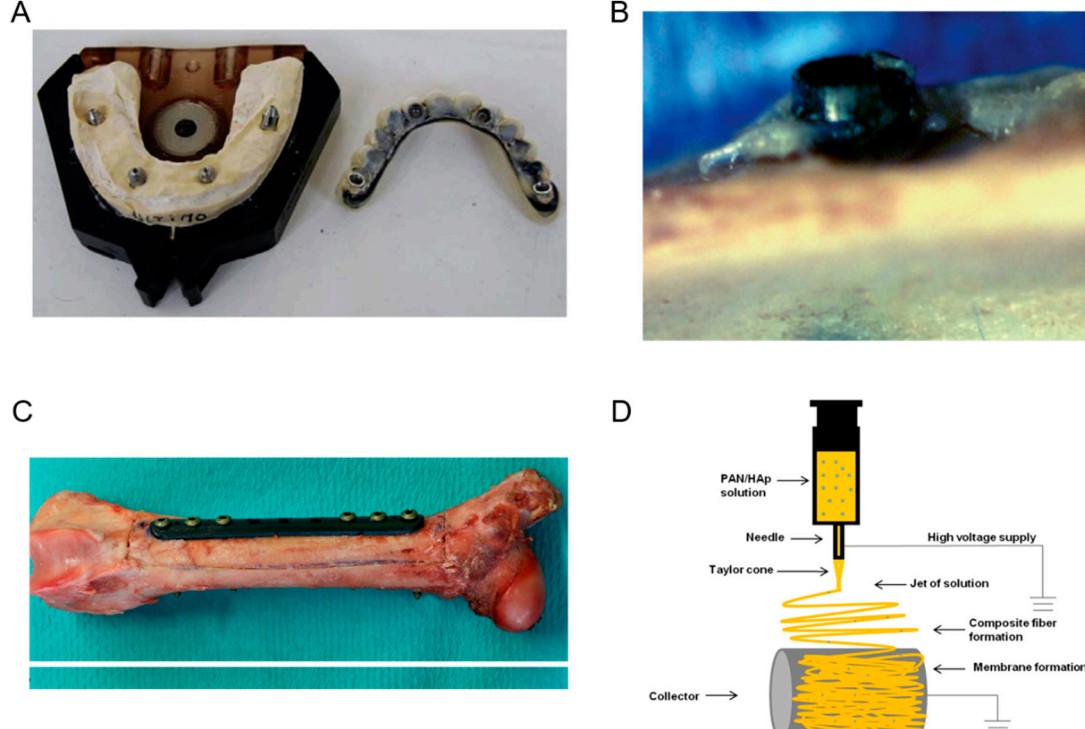

**Figure 8.** (**A**) Dental implant made of CF composite. Reproduced with permission from [69]. (**B**) Pictorial image of the Bisphenyl-polymer/CF implant and tissue. Reproduced with permission from [70]. (**C**) Pictorial representation of the bone prepared with the Carbon Fiber Polyether Ether Ketone (CF/PEEK) implant. Reproduced with permission from [71]. (**D**) Schematic of the electrospinning apparatus for generating CNF. Reproduced with permission from [72].

Figure 9 below shows the differential scanning calorimetry heat flow thermographs of the CF-based nanocomposite, the Alkaline Phosphatase activities of MG-63 cells cultured on different implant materials, the histopathology of the newly formed bone in a rabbit and the computer aided designs of the 3D composite implant.

In another application of CF in dentistry, Pesce et al. evaluated the mechanical characteristics of multi and uni-directional CF structures. They found that the dynamic elastic modulus was higher for the uni-directional fibers while the static elastic modulus was higher for the multi-directional fibers [79]. In a unique application, Wan et al. fabricated 3D CNF scaffolds with bacterial cellulose as the starting source [80]. The 3D composite is made up of CNF and HAp. The produced CNFs had diameters in the range of 10–20 nm and they showed high biocompatibility during in vitro tests. Deng et al. prepared biocompatible and mechanically strong PEEK/n-HA/CF composites for dental implants [81]. The ternary composite shows excellent in vivo bioactivity and promotes fast and effective osseointegration with canine tooth defects. Chen et al. prepared electrically conductive scaffolds from CF for tissue engineering of electroactive tissues [82]. Upon in vitro tests on nerve cells, they found that electrical stimulation promoted cell proliferation and differentiation. Araoye et al. examined the role of CF intramedullary nails in hindfoot fusion [83]. They found that the implant was quite biocompatible and could be safely used for foot and ankle surgery without any considerable risk of failure or further complications.

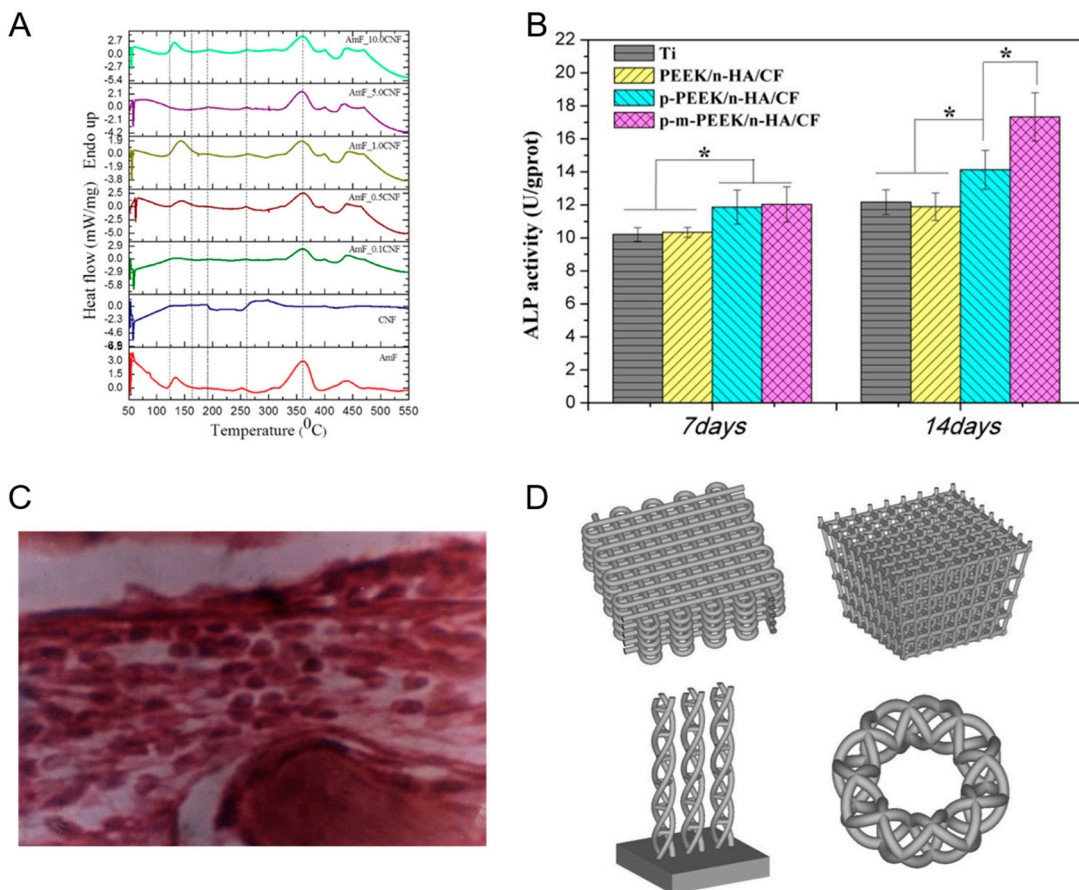

**Figure 9.** (**A**) Differential scanning calorimetry heat flow thermographs of the CF-based nanocomposite. Reproduced with permission from [74]. (**B**) Alkaline Phosphatase activities of MG-63 cells cultured on different implant materials. Reproduced with permission from [75]. (**C**) Histopathology of the newly formed bone in a rabbit. Reproduced with permission from [76]. (**D**) Computer aided designs of the 3D composite implant. Reproduced with permission from [78]. * Represents $p < 0.05$.

## 5. Conclusions

The aviation industry, automotive sector and even sporting goods, all need durable yet light-weight materials so that it can have enhanced performance characteristics. Carbon fiber has contributed enormously to these areas as well as other frontline domains of engineering and medicine. Nano CF reinforced composites have been applied in diverse areas since they display high thermal and electrical conductivity, and superior tensile and compressive strengths. In this short review, we briefly discuss some recent developments in the fabrication of new-age nano CF hybrid materials that have been applied in the fields of sensing, renewables (energy storage systems like batteries and supercapacitors, and solar cells), and tissue engineering. We focus on the working principle of the sensors/renewable devices and how hybridized nano CF alters and boosts the device output characteristics because of superior electrical (increased conductivity) and mechanical (enhanced tensile and compressive strengths) properties.

**Author Contributions:** C.M.D. and L.K. prepared the main draft of the manuscript. G.Y., D.T. and K.-T.Y. helped in revising the manuscript. All authors have read and agreed to the published version of the manuscript.

**Funding:** This work was supported by the Singapore National Research Foundation (NRF) and French National Research Agency (ANR), grant number (NRF2017–ANR002 2DPS). D.T. is grateful for support from the Nanjing Forestry University Startup Fund (163030164).

**Conflicts of Interest:** The authors declare no conflict of interest.

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
