# Peer review of "Multifaceted Hybrid Carbon Fibers: Applications in Renewables, Sensing and Tissue Engineering"

_jcs, doi:10.3390/jcs4030117_

Round 1
Reviewer 1 Report
The Authors presented a Review on the recent work using CF and CNF. It is what the community is working upon at the moment so the topic is interesting and justified for publication. However, some issues should be addressed before publication:
1. The second half of the abstract and the last paragraph of Section 1 make the Authors feel this review is talking about CNF, instead of hybrid, so it should be modified.
2. The second paragraph of Section 2.1 doesn't seem to be related to either CNF or CF.
3. The second half of Abstract, and Section 5 and 6 seem to focus on mechanical properties of CF and CNF in composites, which is not the topic of this Review. The Authors are suggested to make appropriate changes and state what the attributes of CF and CNF for the applications in renewable, sensing and tissue engineering.
4. Many important references are missing, especially in the discussion of energy storage.
5. There are still some typos.
Author Response
1. The second half of the abstract and the last paragraph of Section 1 make the Authors feel this review is talking about CNF, instead of hybrid, so it should be modified.
Answer: The relevant changes have been made in the revised manuscript.
2. The second paragraph of Section 2.1 doesn't seem to be related to either CNF or CF.
Answer: The detailed description of working mechanism of CF-based Vanadium RFB (reference 16) has been changed to reference 17, which is also another type of Vanadium RFB using hybrid CF.
3. The second half of Abstract, and Section 5 and 6 seem to focus on mechanical properties of CF and CNF in composites, which is not the topic of this Review. The Authors are suggested to make appropriate changes and state what the attributes of CF and CNF for the applications in renewable, sensing and tissue engineering.
Answer: The relevant changes have been made in the manuscript.
4. Many important references are missing, especially in the discussion of energy storage.
Answer: Some more energy storage applications of hybrid CFs in fuel cells have been included as a new section 2.4.
5. There are still some typos.
Answer: The revised manuscript has been checked carefully for English language errors and has been corrected accordingly.
Reviewer 2 Report
The review manuscript focuses on the use of modified / multifaceted hybrid carbon fibers in several different scientific areas. Several different examples are provided by the authors and are briefly discussed in order to outline the current advancements and state of art in the use of such fibers for sensing, tissue engineering and renewable devices including storage systems like batteries and supercapacitors, and solar cells. The review is mainly focused on providing the example research and commenting on the positive outcomes of each of them.
The subject of this review is really interesting for the audience of the journal and it is also of interest for the scientific community. The review is not exhaustive, but it highlights several important results relevant to the selected topic. The text is also written in proper English language and it is easy to follow. However, there are several aspects that need to be considered, some of them are general and of increased importance and some others are minor corrections to the text, as listed below.
1. The most important aspect is that the abstract does not properly reflect the content of the document. In more detail, the abstract states the following: “…in this short review, we take a look into the dexterous characteristic of CNF and delve deeply into some of the recent application advancements made by researchers where they have deployed CNF for sensing, tissue engineering and modification of renewable devices.”
However, after reading the whole manuscript it seems that there are too many examples of carbon fiber systems (epoxy composites or other composites) that might be modified by nanomaterials but do not relate with carbon nanofibers (CNFs). For instance, just from the introduction the following are not relevant to CNFs:
- “Ulus et. al. prepared hybrid nanocomposites made of CF, Boron Nitride Nano Particles (BNNP) and CNT…” in lines 52-58,
- “Gabr et. al. added nano-clay as filler material into CF polymer composites in order to improve the strength….” in lines 62-66,
- “Additionally, Zhang et. al. prepared a ternary biocomposite comprising of nano-67 hydroxyapatite/polyamide66 (HA/PA) and CF…” in lines 67-72,
- All the examples in lines 73-92 and 104-110,
and the same stands for the rest of the sections in the manuscript. Therefore, I suggest to the authors to consider rewriting the abstract as well as the last paragraph of the introduction (lines 122-126) in order to reflect more properly the content of the review and not be misleading. A proper description is already included in lines 565-570 of the conclusions.
2. The introduction shall be reconsidered in order to be more clear in what it wants to describe. At its current form it contains a first descriptive paragraph which is very well written and then several paragraphs with examples from literature that are not categorised in any way in order to guide the reader. It is not interesting to read only examples if they are not included in a general context in order to keep the attention of the reader in following the flow of thoughts of the authors.
3. The challenges and alternatives section shall also be reconsidered, as the provided challenges are not relevant to the examined applications (sensing, tissue engineering and renewable devices) but are general for CFRPs. Additionally, and more important is that the provided literature in this section is both outdated and not stemming from scientific sources. The machining of CFRPs is a problem that has been extensively studied and has been overcame in a great extent, i.e. this review considers the drilling and machining technologies and details for CFRPs https://www.sciencedirect.com/science/article/pii/S1359835X1930301X .
4. Throughout the review manuscript, it would be valuable if some conclusions could be added as a paragraph and not a single general sentence in each section.
Also, some indicative minor corrections are:
Line 31: a space is missing between “novel” and “physical” in “novelphysical”
Line 33: There is an additional “a” at the end of the line
Line 72: “CFs show high thermal….” should be moved to a new paragraph.
Line 85: “However, the addition of nano Zirconia…”, The word “However” does not fit at this sentence, consider revising.
Line 95: “…and weak bonding with the bonding.”, it is not clear what the authors mean, consider revising.
Lines 99-100: “They proved that nanoparticle addition into CF composites can significantly boost their performance.” This statement is very general. The boost on the performance is dependent on the type and amount of the nanoparticles, as well as on the method that is used to integrate the nanoparticles in the composite. Consider mentioning the type of nanoparticles and not generalising.
Line 109: “…and lead to matrix…” shall be and “…and led to matrix…”
Lines 111-112: The article “the” is missing before “apparatus” and before “nano sheets”.
Articles are also missing in several of the paragraphs that describe the figures that are included in the manuscript.
Line 129: “…we shall discuss…” shall be replaced by “…we will discuss…”.
Line 176: “…slow oxygen reduction…”, the article “the” is missing.
Line 204: “…have grown enormously since its inception into the market because of its low cost…” shall be “…have grown enormously since their inception into the market because of their low cost…”
Line 302: “…layers in the fiber surface…” shall be “…layers on the fiber surface…”
Line 337: “…with that of Pt.” the word “with” shall be omitted.
Line 373: “…CF-based sensor was due to more number of…” would be better replaced with “…CF-based sensor was due to the increased…”.
Line 443: “…increased the effective surface area because of which there was enhanced mass transport.” …” would be better replaced with “…increased the effective surface area leading to enhanced mass transport.”.
Line 455: The first “and” shall be omitted.
Line 465: In “limbs/other” replace the italic score with “and”.
Line 469: “….thermal expansivity…” shall be “…thermal expansion…”.
Line 512: In “In an altogether different study, BB et. al. used” the last name of the author is PROKIJ not BB.
Line 546: There is a misspelling in the sentence at “… which. C have long…”, remove “.C”
Author Response
- The most important aspect is that the abstract does not properly reflect the content of the document. In more detail, the abstract states the following: “…in this short review, we take a look into the dexterous characteristic of CNFand delve deeply into some of the recent application advancements made by researchers where they have deployed CNF for sensing, tissue engineering and modification of renewable devices.”
However, after reading the whole manuscript it seems that there are too many examples of carbon fiber systems (epoxy composites or other composites) that might be modified by nanomaterials but do not relate with carbon nanofibers (CNFs). For instance, just from the introduction the following are not relevant to CNFs:
- “Ulus et. al. prepared hybrid nanocomposites made of CF, Boron Nitride Nano Particles (BNNP) and CNT…” in lines 52-58,
- “Gabr et. al. added nano-clay as filler material into CF polymer composites in order to improve the strength….” in lines 62-66,
- “Additionally, Zhang et. al. prepared a ternary biocomposite comprising of nano-67 hydroxyapatite/polyamide66 (HA/PA) and CF…” in lines 67-72,
- All the examples in lines 73-92 and 104-110,
and the same stands for the rest of the sections in the manuscript. Therefore, I suggest to the authors to consider rewriting the abstract as well as the last paragraph of the introduction (lines 122-126) in order to reflect more properly the content of the review and not be misleading. A proper description is already included in lines 565-570 of the conclusions.
Answer: This has been taken care of in the revised manuscript.
- The introduction shall be reconsidered in order to be more clear in what it wants to describe. At its current form it contains a first descriptive paragraph which is very well written and then several paragraphs with examples from literature that are not categorised in any way in order to guide the reader. It is not interesting to read only examples if they are not included in a general context in order to keep the attention of the reader in following the flow of thoughts of the authors.
Answer: This has been taken care of in the revised manuscript.
- The challenges and alternatives section shall also be reconsidered, as the provided challenges are not relevant to the examined applications (sensing, tissue engineering and renewable devices) but are general for CFRPs. Additionally, and more important is that the provided literature in this section is both outdated and not stemming from scientific sources. The machining of CFRPs is a problem that has been extensively studied and has been overcame in a great extent, i.e. this review considers the drilling and machining technologies and details for CFRPs https://www.sciencedirect.com/science/article/pii/S1359835X1930301X .
Answer: This has been taken care of in the revised manuscript.
- Throughout the review manuscript, it would be valuable if some conclusions could be added as a paragraph and not a single general sentence in each section.
Answer: This has been taken care of in the revised manuscript.
Also, some indicative minor corrections are:
Line 31: a space is missing between “novel” and “physical” in “novelphysical”
Line 33: There is an additional “a” at the end of the line
Line 72: “CFs show high thermal….” should be moved to a new paragraph.
Line 85: “However, the addition of nano Zirconia…”, The word “However” does not fit at this sentence, consider revising.
Line 95: “…and weak bonding with the bonding.”, it is not clear what the authors mean, consider revising.
Lines 99-100: “They proved that nanoparticle addition into CF composites can significantly boost their performance.” This statement is very general. The boost on the performance is dependent on the type and amount of the nanoparticles, as well as on the method that is used to integrate the nanoparticles in the composite. Consider mentioning the type of nanoparticles and not generalising.
Line 109: “…and lead to matrix…” shall be and “…and led to matrix…”
Lines 111-112: The article “the” is missing before “apparatus” and before “nano sheets”.
Articles are also missing in several of the paragraphs that describe the figures that are included in the manuscript.
Line 129: “…we shall discuss…” shall be replaced by “…we will discuss…”.
Line 176: “…slow oxygen reduction…”, the article “the” is missing.
Line 204: “…have grown enormously since its inception into the market because of its low cost…” shall be “…have grown enormously since their inception into the market because of their low cost…”
Line 302: “…layers in the fiber surface…” shall be “…layers on the fiber surface…”
Line 337: “…with that of Pt.” the word “with” shall be omitted.
Line 373: “…CF-based sensor was due to more number of…” would be better replaced with “…CF-based sensor was due to the increased…”.
Line 443: “…increased the effective surface area because of which there was enhanced mass transport.” …” would be better replaced with “…increased the effective surface area leading to enhanced mass transport.”.
Line 455: The first “and” shall be omitted.
Line 465: In “limbs/other” replace the italic score with “and”.
Line 469: “….thermal expansivity…” shall be “…thermal expansion…”.
Line 512: In “In an altogether different study, BB et. al. used” the last name of the author is PROKIJ not BB.
Line 546: There is a misspelling in the sentence at “… which. C have long…”, remove “.C”
Answer: The above typographical errors have been taken care of in the revised manuscript.
Reviewer 3 Report
Carbon fiber (CF) is one of the most important carbon materials and has received wide attention. In this manuscript, the authors reviewed recent developments of CFs for different applications including renewables, sensing and tissue engineering. The review topic of this manuscript is interesting, but this manuscript is not comprehensive and miss some important parts of this fantastic material. Thus, I recommend to accept the manuscript after a major revision. The comments that the authors should address are as follows:
- The introduction part of the manuscript should focus on the history and definition, importance of the CFs. The synthesis and preparation of CFs should be a separate part.
- Recently, many new methods have been reported to prepare carbon nanofibers. I suggest the authors to include these works in the reviews.
- The physical properties should be in summarized and detailed discussed in a separate part.
- When the author review the CF’s applications in different fields, the authors should demonstrate the uniqueness of CFs compared to other carbon-based materials, such as carbon nanotube and graphene.
- If possible, the authors can outlook more interesting applications of CFs.
Author Response
- The introduction part of the manuscript should focus on the history and definition, importance of the CFs. The synthesis and preparation of CFs should be a separate part.
Answer: A para has been added for this in the revised manuscript.
- Recently, many new methods have been reported to prepare carbon nanofibers. I suggest the authors to include these works in the reviews.
Answer: Some methods have been added in Introduction.
- The physical properties should be in summarized and detailed discussed in a separate part.
Answer: A para has been included in the Introduction dedicated to physical properties.
- When the author review the CF’s applications in different fields, the authors should demonstrate the uniqueness of CFs compared to other carbon-based materials, such as carbon nanotube and graphene.
Answer: The advantages of CF over CNT and Graphene has also been included in the Introduction.
- If possible, the authors can outlook more interesting applications of CFs.
Answer: A new section 2.4 has been added that summarizes fuel cells based on hybrid CF.
Round 2
Reviewer 2 Report
The manuscript has been carefully revised by the authors. The only aspect that has not been considered is the addition of more extensive conclusive paragraphs at the end of each section.
Reviewer 3 Report
The quality of manuscript have been improved after revisions. Now, I can recommend to accept the paper in present form.